

# 30 years of European Commission Radioactivity Environmental Monitoring Database (REMdb) – an open door to boost environmental radioactivity research

Marco Sangiorgi[1], Miguel A. Hernández Ceballos[1], Giorgia Iurlaro[2], Giorgia Cinelli[1], Marc de Cort[1]

[1]European Commission, Joint Research Centre (JRC), Ispra, Italy.
[2]ENEA, Ispra, 21027, Italy

*Correspondence to*: Marco Sangiorgi  (marco.SANGIORGI@ec.europa.eu)

**Abstract.** The Radioactivity Environmental Monitoring data bank (REMdb) was created in the aftermath of the Chernobyl accident (1986) by the European Commission (EC) – DG Joint Research Centre (DG JRC), sited in Ispra (Italy). Since then it has been maintained there with the aim to keep a historical record of the Chernobyl accident and to store the radioactivity monitoring data gathered through the national environmental monitoring programs of the Member States (MSs). The legal basis is the Euratom Treaty, Chapter III Health and Safety, Articles 35 and 36, which clarifies that MSs shall periodically communicate to the EC information on environmental radioactivity levels. By collecting and validating this information in the REMdb, JRC supports the DG for Energy in its responsibilities in returning qualified information to the MSs (competent authorities and general public) on the levels of radioactive contamination of the various compartments of the environment (air, water, soil) on the European Union scale. The REMdb accepts data on radionuclide concentrations from EU MSs in both environmental samples and foodstuffs from 1984 onwards. To date, the total number of data records stored in REMdb exceeds five million, in this way providing the scientific community with a valuable archive of environmental radioactivity topics in Europe. Records stored in the REMDdb are publicly accessible until 2006 through an unrestricted repository "REM data bank - Years 1984-2006" http://doi.org/10.2905/jrc-10117-10024 (De Cort et al., 2007). Access to data from 2007 onwards is granted only after explicit request, until the corresponding monitoring report is published. Each data record contains information describing the sampling circumstances (sampling type, begin-end time), measurement conditions (value, nuclide, apparatus, etc.), location and date of sampling and original data reference. In this paper the scope, features and extension of the REMdb are described in detail.

## 1 Introduction

Radiation occurs when energy is emitted by a source and then travels through a medium, until it is absorbed by matter. In this sense, radiation is a fact of life as every person, animal and object is subjected to radiation every day. Radiation can be either ionizing or non-ionizing. Whenever we refer to radiation in this paper, we mean ionising radiation unless we say otherwise. By definition (ICRP, 1990), ionizing radiation is a radiation with enough energy to break chemical bonds; this



includes X rays and gamma rays, while a source is anything which may cause exposure, such as by emitting ionizing radiation or by releasing radioactive substances or radioactive material, and can be treated as a single entity for purposes of protection and safety.

About 80% of radioactivity in the environment derives from natural sources (background radiation), being the rest associated with the creation and use of artificial sources by human activities, such as medical applications. The main natural sources are: cosmic radiation which is generated by interaction of primary radiation (mainly protons) with atoms of the atmosphere (e.g. Cinelli et al., 2017); terrestrial radiation which is produced as a consequence of the presence of radioactive materials in the Earth's crust, such as $^{40}K$, $^{238}U$ and $^{232}Th$ and their daughter nuclides (UNSCEAR, 2008), and internal radiation, which is generated by the presence of $^{40}K$, $^{14}C$ and $^{210}Pb$ in human bodies from birth. This natural background radiation is enhanced by nuclear accidents, such as Chernobyl and Fukushima (e.g. Imanaka et al., 2015), detonations of nuclear weapons (e.g. Gabrieli et al., 2011), nuclear waste handling and disposal, medical procedures (e.g. diagnostic X-rays, radiation therapy) (e.g. Alkhorayef et al., 2018) and mining (e.g. Carvalho et al., 2014). Nevertheless, radiation caused by artificial radioactive source is no different from natural radiation.

The hazards to people and the environment from radioactive contamination depend on the nature of the radioactive contaminant, the level of contamination, and the extent of the spread of contamination (e.g. Ogundare and Adekoya 2015). Elevated levels of radioactive elements in the environment, resulting from natural or anthropogenic activities, can be a significant problem for the ecosystem and may threaten human health, especially if they build up in the food chain. Consequently, ionising radiation has to be assessed and, if necessary, controlled. In this line, the importance of an adequate characterization of the variability and uncertainty in exposure assessments for human health risk assessments has been emphasized by several national and international organizations (e.g.FAO/WHO, 2006; InVS and AFSSET, 2007). Monitoring radioactivity in the environment is of utmost importance in order to observe trends over time and to verify that there is compliance with the Basic Safety Standards (World Health Organization, 1994).

The transboundary nature and the amount of the contamination during the Chernobyl accident triggered the international organisations to promote international cooperation and communication in nuclear, radiation and emergency preparedness and response. Under the EURATOM treaty (Council Decision 87/600/EURATOM of 14 December 1987), article 36 requires the competent authorities of each Member State (MS) to provide regularly the environmental radioactivity monitoring data resulting from their Article 35 obligations to the European Commission (EC) in order to keep EC informed on the levels of radioactivity in the environment (air, water, milk and mixed diet) which could affect population.

The continuous communication of environmental radioactivity monitoring data after the Chernobyl accident to the EC from MSs triggered the need to integrate, store and organize them so that the collection of information can be easily accessed, managed and updated. With this purpose, the amount and diversity of environmental radioactivity data received from the MSs have been and are routinely stored in the Radioactivity Environmental Monitoring data bank (REMdb) (https://rem.jrc.ec.europa.eu/RemWeb/), which is managed by the EC Joint Research Centre (JRC) sited in Ispra (Italy) as part of its Directorate General for Energy (DG ENER) support programme. REMdb is the base to annually inform of the



radioactivity levels in the environment in the European Community, as stated in art. 35 - 36 of the Euratom Treaty. Every MS has its own database and they submit just a part of their measurements to REMdb.

Environmental radioactivity databases contribute to scientific knowledge of the processes affecting radionuclides distribution and the sources introducing radioactivity into the environment. They provide critical inputs to the evaluation of the

environmental radionuclide levels at regional and global scale, deliver information on temporal trends of radionuclides levels and identifies gaps in available information, as well as they are used as a basis for the assessment of the radiation doses to local, regional and global human populations and biota. In this sense, REMdb makes accessible and understandable to a wider audience radioactivity measurements made by all MS in air, water, milk and mixed diet in the aftermath of the Chernobyl accident, and brings to the scientific community research opportunities to exploit a unique collection of near 5

millions of environmental radioactivity measurements since 1988.

This paper addresses the scope, features and extensions of REMdb with the intention to provide the scientific community with easy access to these data. Since REMdb is constantly growing, this paper decided to refer to data up to 2016 which seem reasonably stable. The REMdb measurements are public until 2006 (http://data.jrc.ec.europa.eu/collection/id-0117), while the access to the data for the 2007– 2016 period is granted after explicit request. In this paper, REMdb data and

monitoring networks are described in Sect. 2, as well as the applied quality control methodology. Then, Sect. 3 presents the status of REMdb for the sampling media recommended by EC. Finally, data record details and conclusions are reported in Sect. 4 and 5 respectively.

## 2 Data and methods

The primary scope of the REMdb is to provide a unique and single framework for working with environmental radioactivity

data originating from many and diverse sources. REMdb was set-up in 1988 and collects in a harmonized format environmental radioactivity data for environmental samples, foodstuffs and other media from 1984 onwards.

There has been significant effort in updating the REMdb at a frequent, regular basis in a consistent and systematic way to preserve the integrity of information embodied to the database. The database is currently hosted on an internal JRC server under, equipped with the latest version of Oracle 12c RDBMS (Relational Database Management System). The servers are

within the JRC internal firewall. They are centrally managed and adhere to all JRC security guidelines and policies. They are accessible from both custom applications (DST, RemWeb, etc.) and third-party software (SQL Developer) through standard communication protocols like SSH, SFTP, and TCP/IP.  Automatic backups occur on the server.

Figure 1 sketches the process from the collection of samples to the final store of data in REMdb. An important tool in this process is the "REM data submission tool" (REM DST) by which data are sent on a regular basis by the national contact

points of EU28 to the JRC. It is worth pointing out that JRC assists MSs in using REM DST correctly by organising regular training courses (e.g. the 2018 edition was held on 13-14 November)

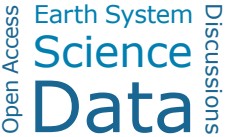

Once gathered the data from MSs, the database is conceived as a series of data records, each one containing a single measurement of a single radionuclide on a single sample, and does not include summary statistics, which have to be created ex post.

Nowadays, the total number of data records stored in REMdb exceeds five million. Figure 2 shows the amount of data stored in REMdb from 1984 to 2016. It is clear that Chernobyl accident generated a peak in the amount of measurements and created consciousness about the importance of monitoring radioactivity. Additionally, the increase of data observed from 1994 to 1996 is also associated with Chernobyl, due to the compilation of contamination data from countries or regions in Europe resulting from the radioactive material released during the accident. For instance, after data validation and inter-comparison, the "Atlas of caesium deposition on Europe after the Chernobyl accident" was published in 1998 (De Cort et al., 1998).

The yearly evolution of measurements stored in REMdb is also influenced, as one can expect, by the amount of countries which joined the REMdb since its creation. The amount of submitted measurements ranges from one million for Germany and France to ten thousand for Ireland, Denmark and Greece. Figure 3 shows the year in which each country started submitting measurements to REMdb, and displays the great difference in the amount of measurements from country to country (in logarithmic scale). Czech Republic, Lithuania, Estonia, Latvia, Poland and Bulgaria (marked in red) joined REMdb since year 2002, but Poland made available samples for year 1986. Slovenia (marked in orange) joined in 2003; Slovakia, and Romania (marked in yellow) joined in 2004; Cyprus (marked in light green) in 2006 and Malta (marked in dark green) in 2008. Even considering this large range of variability of measurements from country to country, the database contains a large and representative number of measurements for each of them.

## 2.1 Quality control

The value of a database system is dependent on the quality of primary data, the organisation and reliability of the data entry and verification systems, and the accessibility of data. It has to be noted that continuous maintenance and frequent updates of the database are also important issues confronted on the practical end.

As part of quality control procedures, all data entered into the database are reviewed and stored once checked and approved by the corresponding data provider. This means that we can avoid spurious or incomplete datasets being available to all users. Further controlled and documented releases of summarised data are timetabled such that values are not changing on a regular basis.

## 2.2 Measurements and Monitoring networks

Measurements carried out by the MSs can be very different from each other's, as they can be referred to different sample types (water, air, food, soil, etc.), different nuclides, different apparatus used for the measurement, different sampling period,



etc. REMdb streamlines the various formats adopted in the EU for reporting routine environmental measurements. In order to bring some harmony and classify the information, some standard sample categories or types are defined and some minimum requirements of the information collected in REMdb are set. Table 1 indicates the information requested in order to be stored as a record in REMdb; not all fields are mandatory. There are three main blocks of information (Locations,

samples and measurements), which mainly refer with source of data and information necessary to describe the procedures for sample collection, treatment, analysis and measurement as performed by the original laboratories (Nweke, et al., 2015).

Itis worth pointing out the attention over field "Less Than" of the measurements record: this field is not mandatory and represents best that can be done with the instruments which vary from country to country. The value is used to indicate that the results of measurements are reported "less than a given value" or "below a threshold detection limit" (called Lower Limit

of Detection). This field is often left empty and no measurements are provided even if they were carried out because considered too low to worth being reported.

A detailed classification of sample types or sample categories in REMdb can be found in (De Cort, et al., 2004). There are eight main categories in this classification: 1) Environmental samples, 2) uncultivated products, 3) crops, vegetable and fruit, 4) manufactured agricultural products, 5) animal products, 6) dairy products, 7) Technical samples and 8) mixed diet. Within

this classification, it is necessary to highlight that sample types can also have subtypes, for example, water samples is subdivided in surface water samples and drinking water samples. Figure 4 shows the amount of single measurements per sample type stored in REMdb. This figure shows how most of sample types have more than a hundred records. One can observe that the most populated category is air-airborne particulate with more than 106 measurements, while a considerable amount of records is also available for the freshwater ecosystem (Surface water and Drinking water). On the contrary, some

sample types and/or subtypes were foreseen but not populated at all, such as Amniotic fluid which belongs to human biological samples.

Nowadays around 70 radionuclides are regularly reported and stored in REMdb. Figure 5 shows the amount of measurements of the most sampled ones stored in the database; amount which varies greatly from year to year, from country to country and also from nuclide to nuclide. In 1996, during the Chernobyl accident, there was a big and sudden increase in

the number of different radionuclides reported and stored in REMdb (almost 100), while in 1994 there was a peak of 110; afterwards it stabilized around 70. In a similar way, some nuclides where of more concern soon after the accident and lost gradually importance due to their short life (e.g. $^{131}$I and $^{134}$Cs), while others gained more and more importance because of their applications in other fields, such as the $^7$Be which is used as an atmospheric tracer (e.g. Lozano et al., 2012). Germany, France and Italy are the countries which report more radionuclides (about 100); UK and Hungary take into consideration

about 60, while the remaining nations focus their attention on 20 radionuclides or even less, which is anyway much higher compared to what recommended by the Commission Recommendation 2000/473/Euratom: gross alfa/beta, $^{137}$Cs, $^3$H, $^{90}$Sr, $^{40}$K, $^{14}$C, $^7$Be.

Figure 6 shows how the main radionuclides stored in REMdb are also those with the largest country coverage. For example, more than 30 countries measure total-beta, $^{137}$Cs and $^7$Be. Looking at this figure, 72% of the total number of radionuclides is



measured in more than 20 countries, and 22% in more than 30 of them. This fact is associated with the number of monitoring stations deployed in each country reporting high-quality measurements to REMdb. In Europe, different approaches in the definition of the national environmental monitoring networks have been adopted, based on socio-economic considerations about where to install monitoring stations.

**2.3 Dense and sparse network**

Figure 7 shows an example of the sampling locations distributed all over Member States' territories in 2006 corresponding to $^{137}$Cs in airborne particulates. Differences in the number of sampling locations between both plots are associated with the definition of dense and sparse monitoring networks (Commission Recommendation 2000/473/Euratom). The sparse network is included in the dense network, being a subset of it. There is no distinction between the structures of data stored in REMdb
as dense or sparse network: both kinds of data comply with the same requirements. While dense network refers to a monitoring network that comprises sampling locations distributed throughout the Member State's territory and used by the Commission to compute regional averages for radioactivity levels in the Community, sparse network groups those locations in which high-sensitivity measurements are performed. A sparse monitoring network, means therefore a monitoring network which comprises for every geographical division and for every sampling medium at least one representative location of that
geographical division.

**3. Status of REMdb as recommended by EC**

Activities such as the medical uses of radiation, the operation of nuclear installations, the production, transport and use of radioactive material, and the management of radioactive waste must be subject to standards of safety (Basic Safety Standards, 2014). The recommendation of 8 June 2000 on the application of Article 36 of the Euratom Treaty
(2000/473/Euratom, EC 2008) concerning the monitoring of the levels of radioactivity in the environment for assessing the exposure of the population as wholes suggests that some sample types and radionuclides are of more concern due to their higher contribution to the annual dose. Table 2 shows the sample types and radionuclide categories to be monitored and reported to the Commission. These radionuclides are therefore measured and reported thoroughly by the MSs, as can be seen in Figure 5.
All the measured radionuclides except $^{90}$Sr and $^{137}$Cs can be of either natural or artificial origin. The two exceptions are of artificial origin, mainly from past atmospheric weapons testing and from radioactive routine or accidental discharges from nuclear facilities.

Gross alpha and gross beta measurements are appropriate to characterise the total radioactivity of the sample. Gross beta is, by definition, the total measured beta activity in a sample; beta from tritium and in general very low energy beta emitters are
normally not considered and short lived radon daughters are excluded through a sufficient delay time (e.g. five days) before counting. In fact, gross beta analysis does not detect weak beta-emitters such as those emitted by $^3$H, $^{14}$C, $^{35}$S and $^{129}$I.





Residual beta is the measured gross beta activity minus $^{40}$K activity, being the latter the main natural source of activity in water.

Following the contamination of the environment with radionuclides, the population is exposed through both external and internal irradiation pathways. From a radio-ecological point of view, the behaviour of radionuclides in the environment and

the interaction (uptake, excretion) with food organisms (plants, fungi, animals) is essential for the prediction of future internal exposure due to intake of contaminated foods (e.g. Merz et al., 2016).

### 3.1 Air

Airborne particulate is measured due to its greater radiological significance. Airborne radioactive materials may occur in either gaseous or particulate form. In general, the latter is of greater potential radiological significance because it may be

deposited and hence remain in the local environment. For instance, regarding emissions from Fukushima, $^{137}$Cs was attached in the size range 0.1−2μm diameter (Kaneyasu,et al., 2012). Consequently, most national routine monitoring networks measure only the particulate component.

Airborne particulate sampling is carried out by pumping air through filters. In most countries filters are changed daily and analysed for total beta activity following the decay of radon decay products. Individual radionuclide analyses are performed

weekly, monthly or quarterly. Figure 8 shows the amount of measurements by country for $^{137}$Cs and $^{7}$Be in the air. Man-made alpha-emitting aerosols are rarely measured by routine monitoring networks as they are usually undetectable, even close to the nuclear installations where they are produced. $^{137}$Cs and $^{7}$Be are normally measured with a gamma spectrography at the same time, therefore the amount of reported measurements for both nuclide should be the same, but it does not happen because of lack of harmonization between countries.

$^{137}$Cs is of major concern because it is the most abundant anthropogenic radionuclide and, because of its high volatility, the radiation type it emits and its biological activity (chemical analogue of potassium). On the other hand, the cosmogenic radionuclide $^{7}$Be is important because of its high contribution to the annual dose. In addition, $^{7}$Be is widely used as aerosol tracer in order to study aerosol transport and removal in the troposphere by testing scavenging parametrizations (e.g. Alonso-Hernandez., 2014).

### 3.2 Water: Surface and drinking

The presence of artificial radionuclides in marine environment results from global fallout from atmospheric nuclear weapons tests, fallout from the Chernobyl accident, discharges of radionuclides from nuclear installations, contributions from nuclear testing sites, past dumping of radioactive wastes, nuclear submarine accidents, loss of radioactive sources, applications of radionuclides in medicine and in industry, and the burn-up of satellites using radionuclides as their power source (Livingston

and Povinec, 2000). They pose a number of health hazards, especially when these radionuclides are deposited in the human body through drinking, and can eventually become incorporated into sediments and living species.



Due to man's activities, the fraction of $^3$H and the presence of $^{137}$Cs have to be checked in this sampling media. Natural radionuclides in river water include $^3$H at levels of [0.02 - 0.1] Bq l$^{-1}$, $^{40}$K [0.04 - 2 Bq l$^{-1}$], radium, radon and their short-lived decay products [< 0.4 - 2 Bq l$^{-1}$] (De Cort, et al., 2004), while $^{137}$Cs is the most abundant anthropogenic radionuclide present in the marine environment (e.g. Aarkrog et al., 1997).

Surface water is one of the compartments into which authorised discharges of radioactive effluents from nuclear installations are made and hence, radionuclides in this sampling media can be either found in the water phase or associated with suspended particles and sediments. For instance, the Chernobyl accident in 1986 contributed significantly mainly to $^{134}$Cs and $^{137}$Cs inventories in seawater of the Baltic and North Seas, resulting in the Baltic Sea being the most highly contaminated by $^{137}$Cs (e.g. Nies and Nielsen, 1996). (Povinec et al., 2003) analyse the distribution of anthropogenic $^{137}$Cs in surface

waters of the NE Atlantic Ocean for the year 2000, reporting mean values from 60±50 Bq m$^{-3}$ for the Irish Sea to 2.1±1.2 Bq m$^{-3}$ for the English Channel.

Samples are either taken continuously and bulked for monthly or quarterly analysis, or alternatively, spot samples are taken periodically several times a year and analysed individually or as a composite. Figure 9a shows the total number of $^{137}$Cs and total beta measurements regarding surface water media in REMdb.

Drinking water, on the other hand, is monitored because of its vital importance for man, even though a severe radioactive contamination of this medium is unlikely. Samples may be taken from ground or surface water supplies, from water distribution networks, mineral waters etc. Spot samples are taken a few times a year and, analysed individually or samples are taken daily and bulked for monthly or quarterly analysis. The number of measurements included in REMdb for this sampling media is shown in Figure 9b. Same as for surface water, most important natural radionuclides in drinking water are

$^3$H, $^{40}$K, radium, radon and their short-lived decay products, but vary greatly. Eventual presence of $^3$H, $^{90}$Sr and $^{137}$Cs and radium may also be due to man's activities.

### 3.3 Milk

For many contamination scenarios, especially for accidentally released radionuclides, consumption of milk and dairy products has been shown to be one of the most important pathways to the internal dose to the public (UNSCEAR, 2000). As

an example of its importance, monitoring of $^{137}$Cs and $^{90}$Sr in consumption milk is ongoing since 1955 and 1960 in Sweden (http://projects.amap.no/project/monitoring-of-cs-137-and-sr-90-in-consumption-milk/) and Finland (http://www.radioecology-exchange.org/content/monitoring-sr-90-and-cs-137-milk-finland respectively)

Figure 10 shows the total number of measurements of $^{137}$Cs, $^{40}$K, and $^{90}$Sr in cow's milk per European country. Samples are mostly taken at dairies covering large geographical areas in order to obtain representative samples. They are generally taken

on a monthly basis; but sometimes only during the pasture season. Generally, the concentrations of the stable elements calcium (Ca) and potassium (K) are determined because of the similarity of their metabolic behaviour with strontium (Sr) and caesium (Cs) respectively.





### 3.4 Mixed diet

The aim of measuring radioactivity in mixed diet is to get "integral" information on the uptake of radionuclides by man via the food chain. Samples are taken as ingredients or as complete meals, mostly at places where many meals are consumed (i.e. factory restaurants, schools). The trend is to sample complete meals according to food consumption statistics to give a

representative figure for the contamination of mixed diet; it's also very common to analyse single foodstuff and combine them according to local diet style. Knowledge of the contamination of the individual ingredients together with the composition of the national diet can also lead to a representative figure.

Rather than expressing the radioactivity content of foodstuffs per unit weight, it is more appropriate to estimate the activity consumed per day per person (Bq $d^{-1}$ $p^{-1}$). Generally, the concentrations of the stable isotopes of calcium and potassium are

determined because of the similarity of their metabolic behaviour with strontium and caesium, respectively. Typical values in mixed diet are 0.7 to 1.5 g $d^{-1}$ person$^{-1}$ for calcium and 3 to 4 g $d^{-1}$ person$^{-1}$ for potassium. Figure 11 shows the total amount of measurements for sample type mixed diet, $^{137}$Cs, $^{90}$Sr and $^{14}$C by country, as recommended in Table 2.

### 4 Data Availability

Environmental radiation monitoring datasets are freely available at JRC Data Catalog

http://data.jrc.cec.eu.int/collection/id-0117 .

Datasets can be downloaded singularly as Excel files and zipped due to their dimension year by year from 1984 to 2006 by following subsequent PIDs:

REM data bank - Year 1984" PID: http://data.europa.eu/89h/jrc-10117-10001

REM data bank - Year 1985" PID: http://data.europa.eu/89h/jrc-10117-10002

REM data bank - Year 1986" PID: http://data.europa.eu/89h/jrc-10117-10003

REM data bank - Year 1987" PID: http://data.europa.eu/89h/jrc-10117-10004

REM data bank - Year 1988" PID: http://data.europa.eu/89h/jrc-10117-10005

REM data bank - Year 1989" PID: http://data.europa.eu/89h/jrc-10117-10006

REM data bank - Year 1990" PID: http://data.europa.eu/89h/jrc-10117-10007

REM data bank - Year 1991" PID: http://data.europa.eu/89h/jrc-10117-10008

REM data bank - Year 1992" PID: http://data.europa.eu/89h/jrc-10117-10009

REM data bank - Year 1993" PID: http://data.europa.eu/89h/jrc-10117-10010

REM data bank - Year 1994" PID: http://data.europa.eu/89h/jrc-10117-10011

REM data bank - Year 1995" PID: http://data.europa.eu/89h/jrc-10117-10012

REM data bank - Year 1996" PID: http://data.europa.eu/89h/jrc-10117-10013

REM data bank - Year 1997" PID: http://data.europa.eu/89h/jrc-10117-10014

REM data bank - Year 1998" PID: http://data.europa.eu/89h/jrc-10117-10015



REM data bank - Year 1999" PID: http://data.europa.eu/89h/jrc-10117-10016

REM data bank - Year 2000" PID: http://data.europa.eu/89h/jrc-10117-10017

REM data bank - Year 2001" PID: http://data.europa.eu/89h/jrc-10117-10018

REM data bank - Year 2002" PID: http://data.europa.eu/89h/jrc-10117-10019

REM data bank - Year 2003" PID: http://data.europa.eu/89h/jrc-10117-10020

REM data bank - Year 2004" PID: http://data.europa.eu/89h/jrc-10117-10021

REM data bank - Year 2005" PID: http://data.europa.eu/89h/jrc-10117-10022

REM data bank - Year 2006" PID: http://data.europa.eu/89h/jrc-10117-10023

or as a bulk "REM data bank - Years 1984-2006 at" http://doi.org/10.2905/jrc-10117-10024) (De Cort et al., 2007).

The fields made available for download are: locality name, country name, locality latitude, locality longitude, sample type description, sample treatment description, sample begin date time, sample end date time, laboratory name, apparatus description, nuclide, measured standard activity value, measured standard unit, value type description. Datasets size can be

up to 35 MB and contain up to two hundred lines, each line representing a single record or measurement.

Users can also access the REMdb on-line and retrieve them through personal queries.  Monitoring reports from year 1996 to 2006 are also available for download at the corresponding dataset.

For full database access or questions, please write an e-mail to JRC-REMDBSUPPORT@ec.europa.eu

## 5 Conclusions

This paper provides a synthesis of the online Radioactivity Environmental Monitoring data bank (REMdb). The database presented here contains more than five million of records from 1984 onward concerning radioactivity levels in Europe of air, deposition, water, milk, meat, crops and vegetables; REMdb brings to the scientific community endless of research opportunities. The database is accessible via a web browser and is open to the scientific community. During its more than 30 years of operational life, REMdb grew more and more in amounts of measurements thanks also to more countries joining the

project. Nevertheless, experience has shown that its structure could be optimized in order to accommodate users' needs and compare measurements carried out by different laboratories. Future work focuses on maintaining the updates frequency constant, enhance the user's accessibility to original data, and develop a user-friendly interface.



**Author contributions**

MDC was in charge of REBdb creation and is the main reference for information being the person responsible of the project; MS is the main author, MAHC was the main reviewer and gave technical guidance, GC and GI gave very valuable contribution with their comments.

**Competing interests**

The authors declare that they have no conflict of interest.

**Disclaimer**

By accessing the REM Database and its content, the user agrees that the law of the European Union, without regard to principles of conflict of laws, will govern these terms and conditions and any dispute of any sort that might arise between the
user and the European Commission. The European Union reserves the right to modify or discontinue temporarily or permanently, the REM Database (or any part thereof) without prior notice to users.

**Acknowledgement**

The authors would like to thank all the EU Member States for having sent the data and David Carlson for his very helpful and kind support.

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



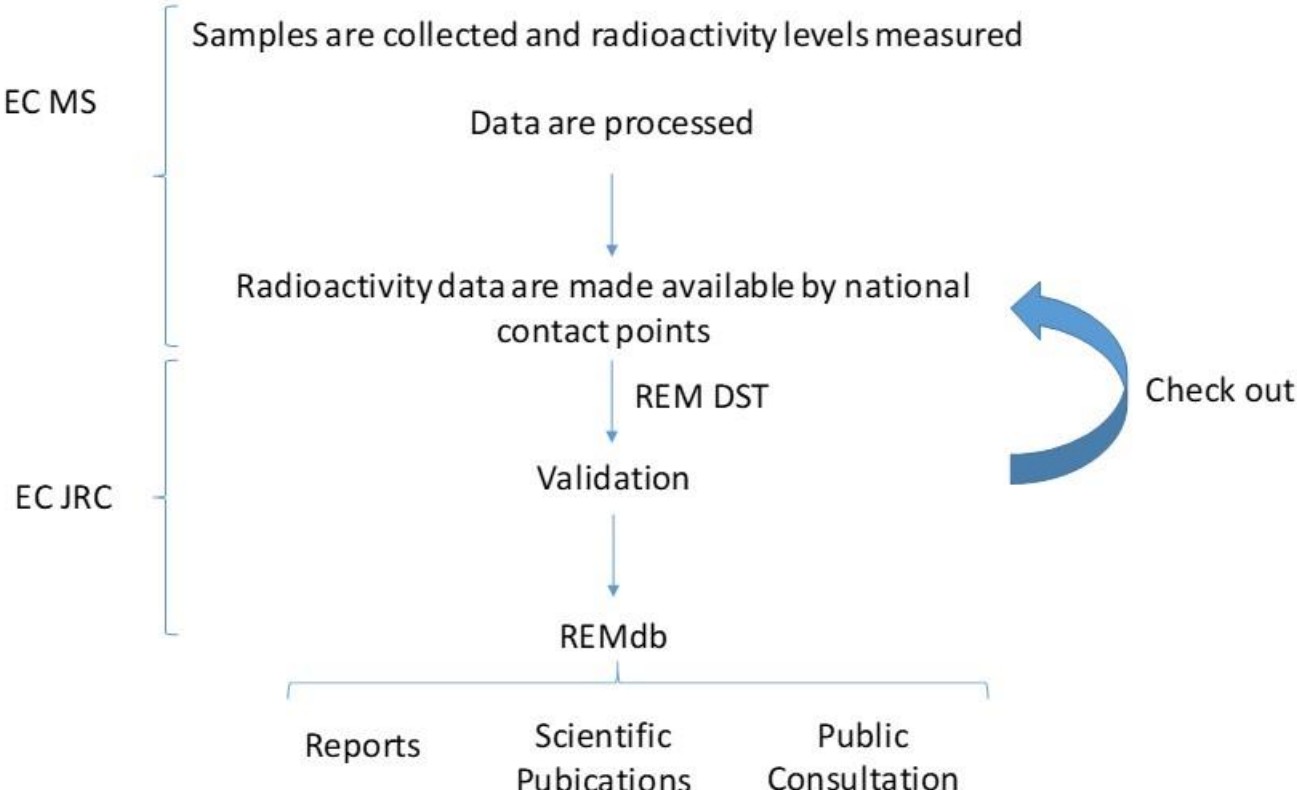

**Figure 1: Flow of data to be stored in the REMdb.**





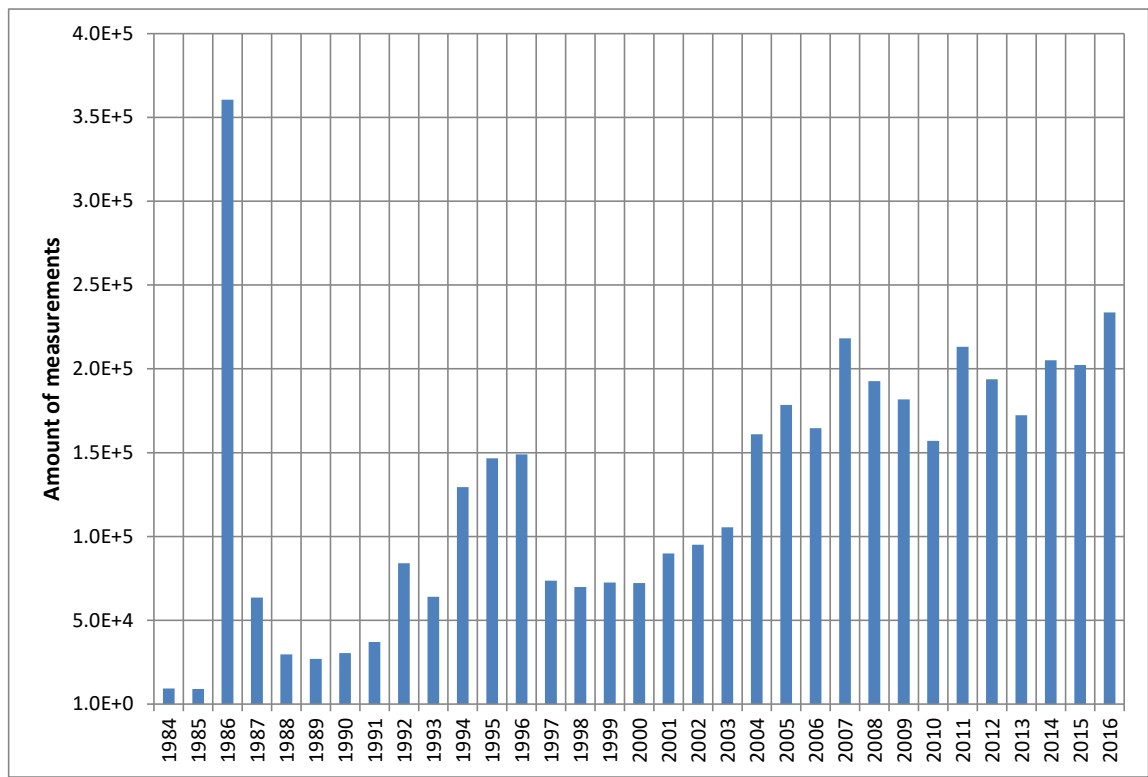

**Figure 2. Amount of single measurements stored in the REMdb. (Please note that even if REMdb is active since 1988, some earlier measurements are also stored)**



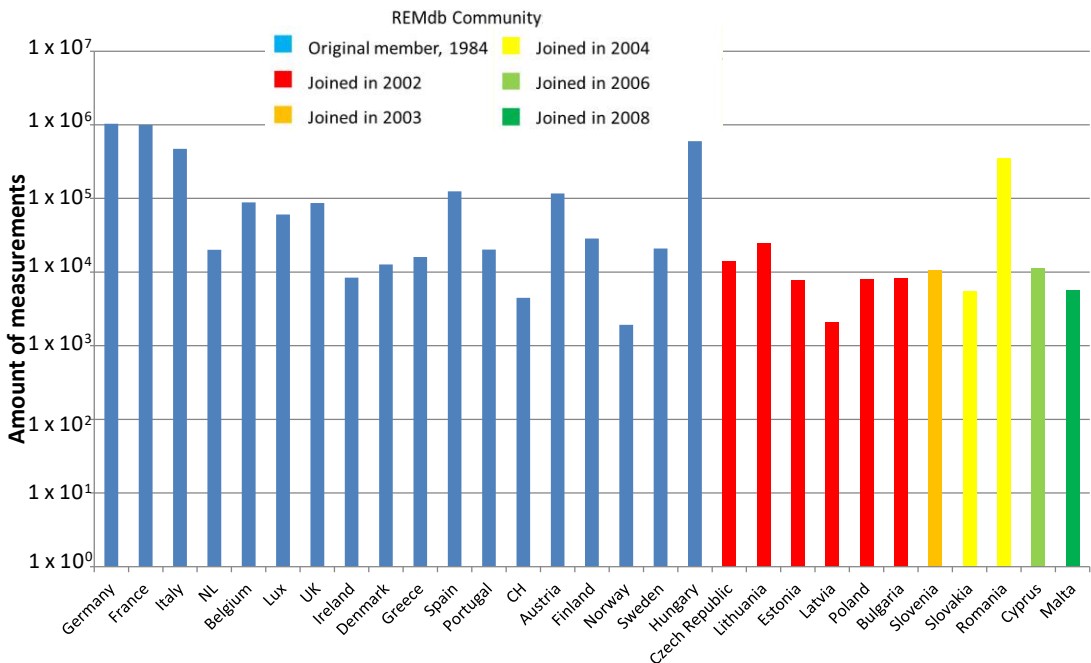

**Figure 3. Country participating to REMdb according to their year of joining. (Malta joined in 2008) and amount of measurements stored in REMdb from each country**

| Locations | | Samples | | Measurements | |
|---|---|---|---|---|---|
| | Name | | Sample Type | | Apparatus |
| | Catchment | | Sample Treatment | | Nuclide |
| | NUTS Code | | Begin Date/Time | | Less Than |
| | Latitude | | End Date/Time | | Activity Value |
| | Longitude | | Laboratory | | Measurement Unit |
| | Height | | | | Value Type |
| | Coordinate Accuracy | | | | Uncertainty Value |
| | | | | | Uncertainty Type |
| | | | | | Uncertainty Unit |
| | | | | | Comment |
| | | | | | Sparse Graph |

5    **Table 1. Mandatory fields associated with each data stored in REMdb**






**Figure 4. Amount of measurements per sample type as specified in REMdb (logarithmic scale, subtypes Surface water and Drinking water are included in type Water samples, subtype Milk-Cow is included in type Cow).**



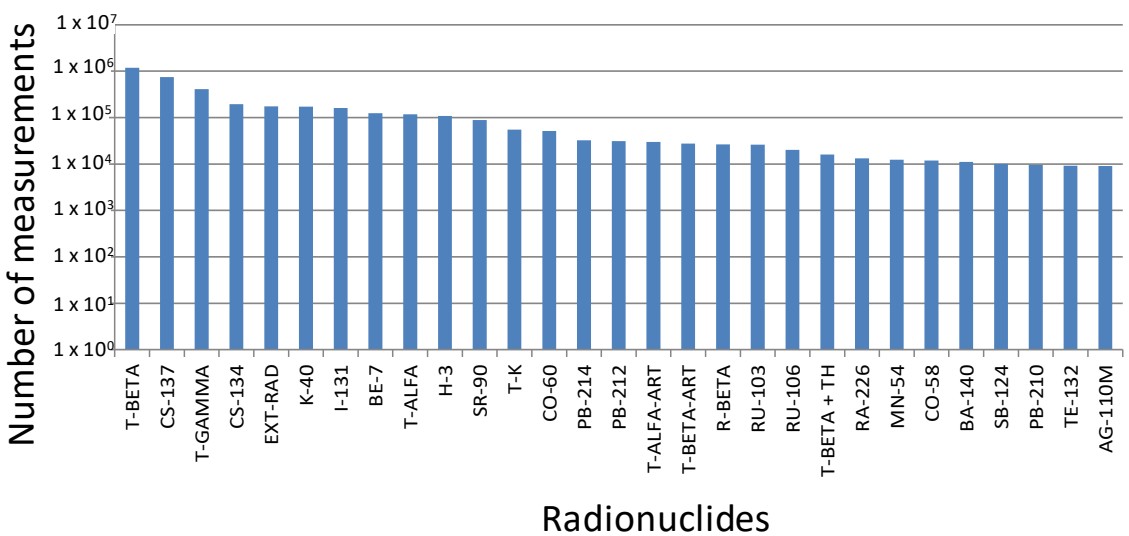

**Figure 5. Amount of single measurements grouped by radionuclide stored in REMdb; the figure shows just those with a number of measurements above 1 x 10⁴.**

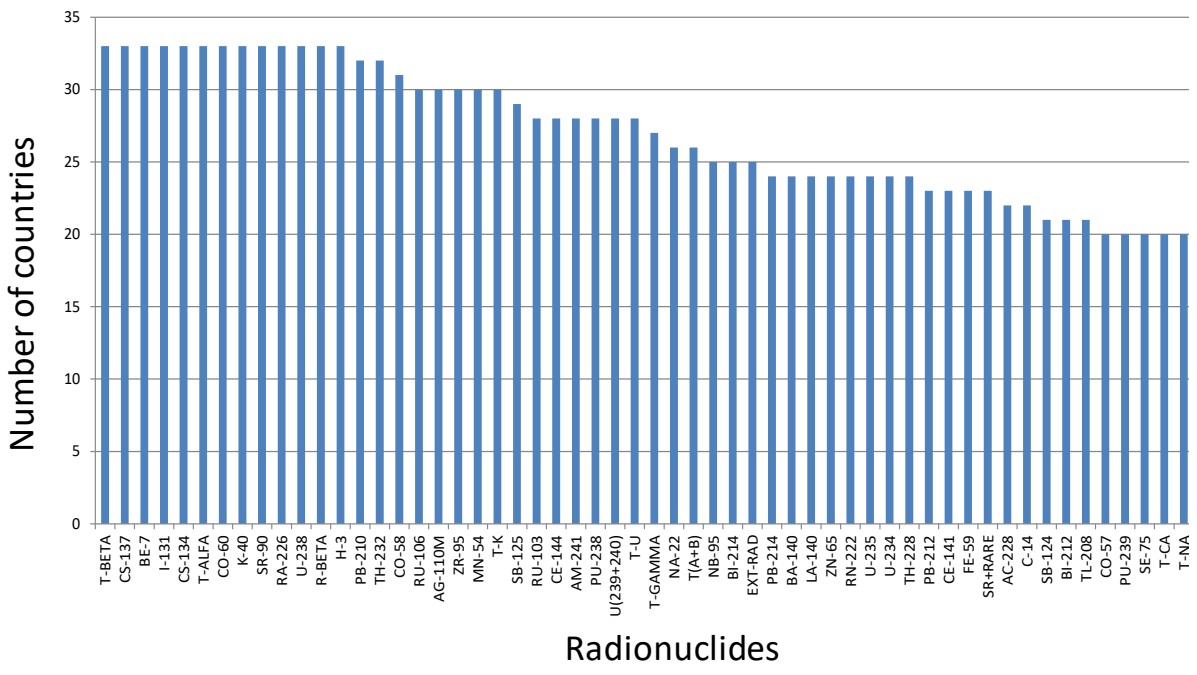

**Figure 6. Number of countries providing information of a certain radionuclide to REMdb.**



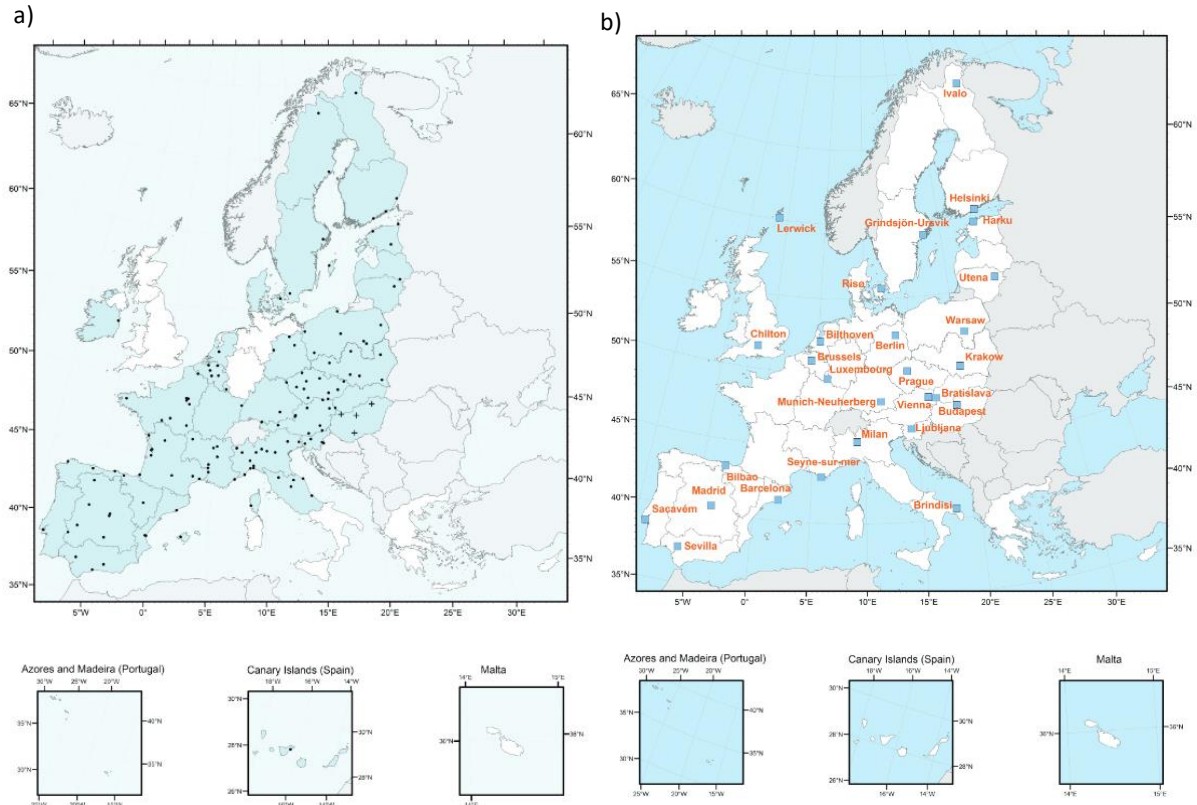

**Figure 7. Example of sampling locations of dense (left) and sparse (right) network for $^{137}$Cs as airborne particulates, year 2006**

| Sampling Media | Radionuclide category | |
| --- | --- | --- |
| | Dense network | Sparse network |
| Airborne particulates | $^{137}$Cs, gross beta | $^{137}$Cs, $^{7}$Be |
| Surface water | $^{137}$Cs, residual beta | $^{137}$Cs |
| Drinking water | Tritium, $^{90}$Sr, $^{137}$Cs | Tritium, $^{90}$Sr, $^{137}$Cs |
| Milk | $^{137}$Cs, $^{90}$Sr | $^{137}$Cs, $^{90}$Sr, $^{40}$K |
| Mixed diet | $^{137}$Cs, $^{90}$Sr | $^{137}$Cs, $^{90}$Sr, $^{14}$C |

**Table 2: Sample types and measurements as recommended in (Basic Safety Standards, 2014)**



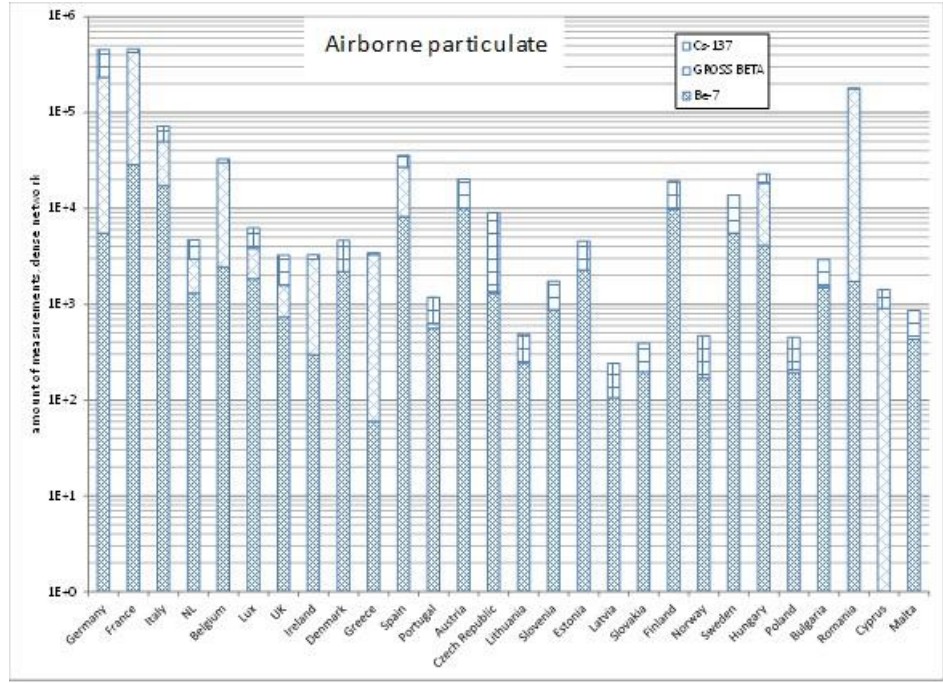

**Figure 8. Total amount of measurements (dense network) for sample type airborne particulates, $^{137}$Cs, total beta and $^{7}$Be, by country as recommended by (2000/273/Euratom).**





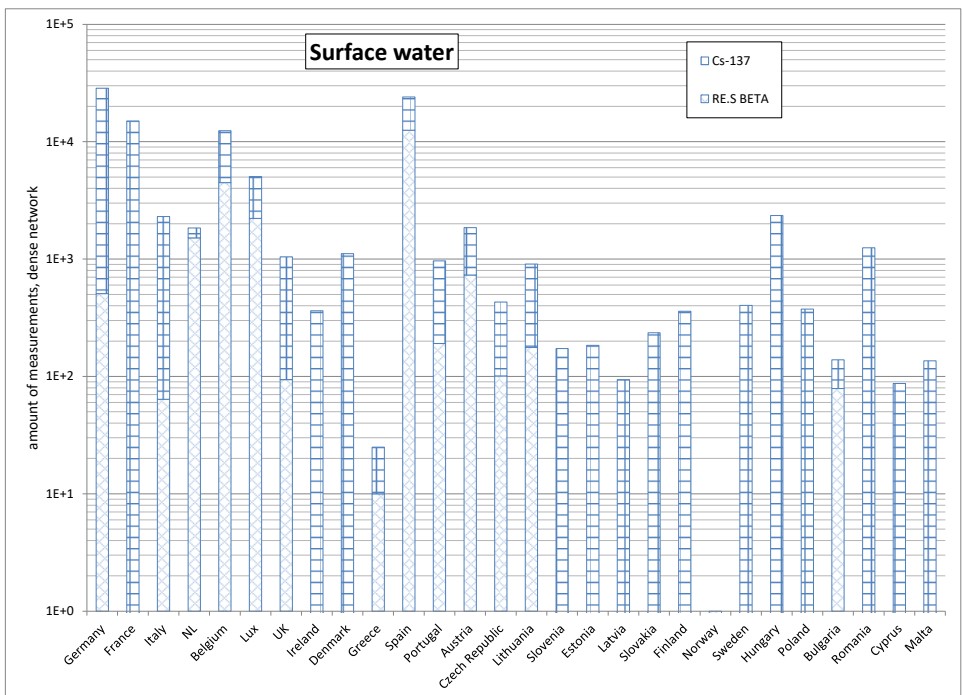

**Figure 9a. Amount of measurements (dense network) for surface water, by country, as recommended in Table 2.**

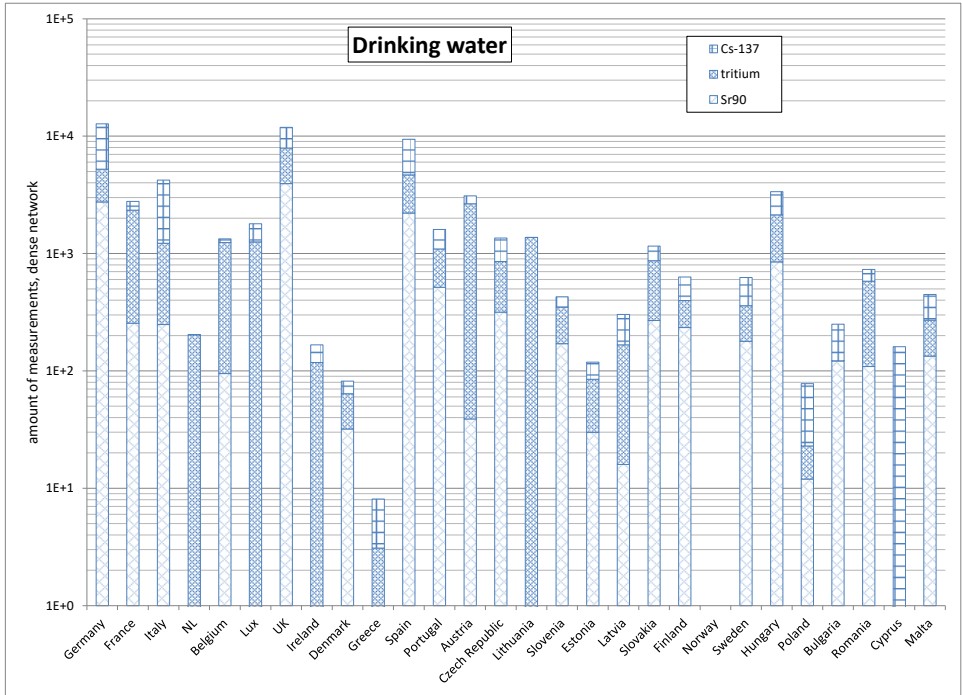

**Figure 9b. Amount of measurements (dense network) for drinking water, by country, as recommended in Table 2.**



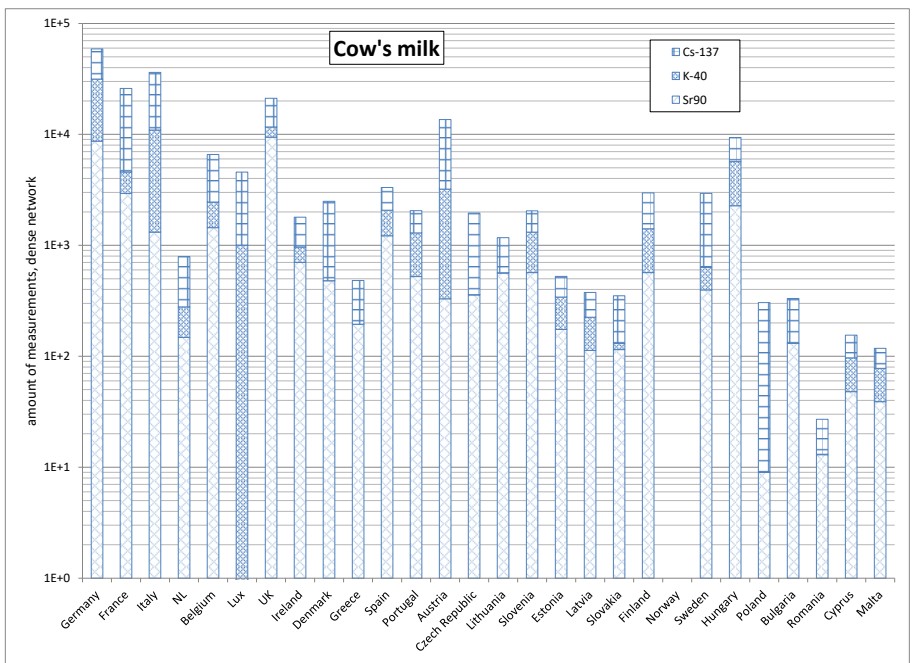

**Figure 10.** Amount of measurements (dense network) for sample type cow's milk, $^{137}$Cs, $^{40}$K and $^{90}$Sr, by country, as recommended in Table 2.

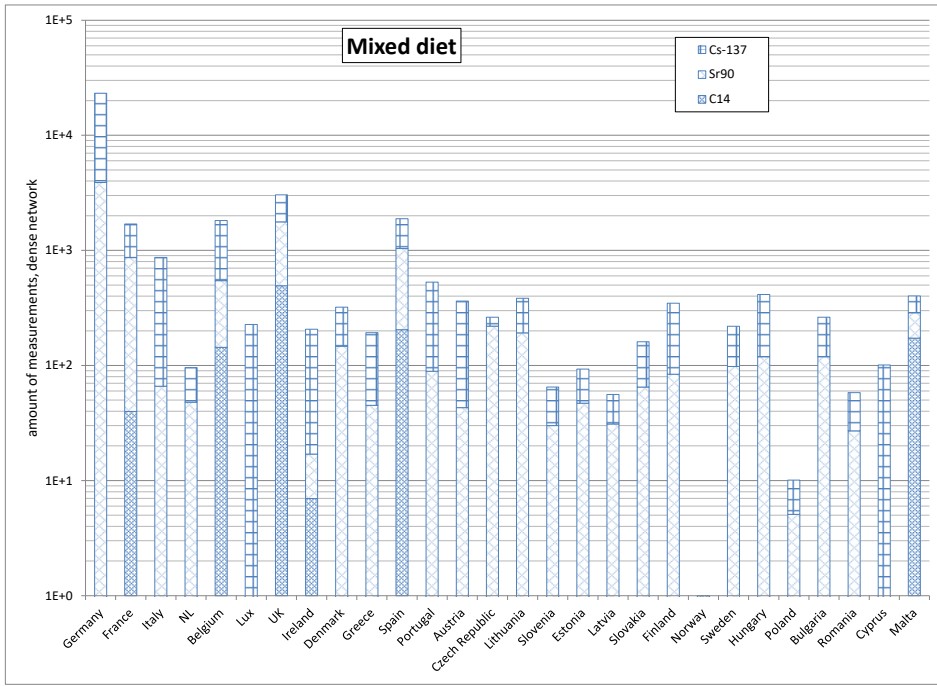

5     **Figure 11.** Amount of measurements for sample type mixed diet, $^{137}$Cs, $^{90}$Sr and $^{14}$C, by country, as recommended in Table 2.