# Peer review of "years of European Commission Radioactivity Environmental Monitoring Database (REMdb) – an open door to boost environmental radioactivity research"

_Earth System Science Data, 2018_

## Referee Comment (RC1) · Anonymous Referee #1 · 13 Feb 2019

Referee comments:

30 years of European Commission Radioactivity Environmental Monitoring Database (REMdb) –an open door to boost environmental radioactivity research by Marco Sangiorgi et al.

The manuscript deals with a database containing radioactivity data from environment, food chains etc. The database data flow relies on the EU member states' authorities that regularly send national data to the European Union. The paper is well written

and it deserves to be published, especially as the existence of REMdb is not that well known. Even I, after 30 years work experience with environmental radioactivity, had never heard of such a resource. I suggest publication of the manuscript in Earth System Science Data once the authors have taken into consideration some minor suggestions found below.

General comments

To put the REMdb to a wider context I wonder if similar more or less public databases are available elsewhere? Are the MS competent authorities the only data providers? University datasets often provide useful information and nowadays the funding organizations often require an open data policy. Are there plans to extend the time period backwards from 1984? Important data was gathered during the period of atmospheric weapons testing.

Detailed comments

Citation on page 2, line 23: Do the authors mean this? International Atomic Energy Agency & World Health Organization. (‎1996)‎. International basic safety standards for protection against ionizing radiation and for the safety of radiation sources. Vienna : International Atomic Energy Agency. http://www.who.int/iris/handle/10665/41593

Page 5, line 24: "In 1996, during the Chernobyl accident, there was. . ." 1986?

Page 6, line 31: "In fact, gross beta analysis does not detect weak beta-emitters such as those emitted by 3H, 14C, 35S and 129I." Maybe the authors should tell that total beta activity results are always dependent on the instrument used. Some instruments can measure even low-energy beta particles.

Page 7, lines 8-12: Maybe the gaseous iodine should also be discussed.

Page 7, Lines 13-14: "In most countries filters are changed daily and analysed for total beta activity following the decay of radon decay products." How about "after the decay

of short-lived radon progeny"?

Page 7, lines 17-19: " 137Cs and 7Be are normally measured with a gamma spectrography at the same time, therefore the amount of reported measurements for both nuclide should be the same, but it does not happen because of lack of harmonization between countries." spectrography -> spectrometry? both nuclide -> both nuclides? Maybe the amount of reported measurements for both nuclides differ also due to "<MDA" values?

Page 7, lines 21-22. Is beryllium-7 significant from dose point of view? If so, please, add a literature reference.

Page 7, lines 25-31: Please, clarify the term "surface water". Does it mean fresh water in lakes and rivers or is also surface water of oceans included? I would expect the radionuclide content of water and intake by drinking to be negligible compared to aquatic food chains ending to man.

Page 8, lines 9-11. Is the high Cs-137 content of ocean water in the Irish Sea due to Sellafield emissions or the Chernobyl accident?

Page 8, lines 20-21: "Eventual presence of 3H, 90Sr and 137Cs and radium may also be due to man's activities." Isn't the presence of Sr-90 and Cs-137 solely due to anthropogenic activities?

---

## Referee Comment (RC2) · Anonymous Referee #2 · 16 Feb 2019

General comments

The manuscript "30 years of European Commission Radioactivity Environmental Monitoring Database (REMdb) – an open door to boost environmental radioactivity research" describes the REMdb database which is a product of a more than a three decade-long radioactivity monitoring effort and collaboration of European member states. The long time span, vast geographical coverage, variety of sample types and the immense number of measurement records result in an invaluable dataset, which will undoubtedly prove of great value for the scientific community. In this light, the

manuscript fits very well into the scope of the journal "Earth System Science Data" and can be considered for publication after the authors address the comments posted below.

The manuscript provides links to yearly and bulk datasets which can be downloaded as Excel files. Data from REMdb can also be accessed by an online query tool where the user can personalise the search by location, sample type, observation period, export format etc. The files on the provided links and the files provided by the online query tool are compliant with the descriptions provided in the Data Availability section.

The manuscript accompanying data does, however, have a major issue which the authors should discuss with the Editor before revision. The present database is composed of two datasets. While the first one spanning between 1984-2006 (De Cort et al., 2007) is compliant with the data policies posted on ESSD websites and further elaborated in a recent Editorial (Carlson and Oda, 2018), the second dataset (2007-2016) is not. Namely, it does not have a DOI nor is it fully publicly available (explicit request by email is needed for access; P10 L18). Additionally, the part of the Disclaimer in P11 L10-11 ("The European Union reserves the right to . . . discontinue temporarily or permanently, the REM Database. . .") could prove controversial regarding the above mentioned data policies of ESSD.

Specific comments

In P1 L9 the DG abbreviation is not explained.

P3 L10 and P1 L17: The abstract says the database contains measurements since 1984, while in page 3 it says since 1988

P7 L15 and Fig. 8: "Figure 8 shows the amount of measurements by country for 137Cs and 715 Be in the air." For unambiguity the authors should clarify that this refers to the total amount of measurements in the database.

P7 L26: "aquatic" is probably more appropriate than "marine"?

[Figure]

Section 4: The "Data availability" section should include procedure for data after 2006, i.e. it should be explicitly stated in P10 L18 that the full database also contains measurements after 2006. Additionally, I suggest the authors do not only mention, but also include a short description of the REMdb online query tool and its functionality as it offers useful search options and additional export formats which many readers could find beneficial.

P10 L11-14: The abbreviations used in the Excel files should be mentioned in the paper, for example: "locality name (LOC_NAME),..., apparatus description (APT_DESCRIPTION), nuclide (NUC_CODE)..."

Figure design of the graphs in the manuscript is variable, for example: some have a frame (Figs. 2, 8-11) and some do not (Figs. 3-6); font sizes of axis titles in Figs. 5 and 6 are much larger compared to similar graphs in the manuscript.

Fig. 7 shows the sampling distribution from 13 years ago. As the authors present the database up until 2016, a more recent picture would be appropriate.

Fig. 8: The legend in the figure is so small that the reader cannot see which symbols are used for 137Cs, total beta and 7Be

Fig. 9a: Again the legend is too small to recognise the symbols of the radionuclides

Figs. 9a and 9b: There should be only one subscript per figure

Figs. 8-11: I suggest to add "in REMdb" to avoid ambiguity (e.g. "Total amount of measurements in REMdb (dense network) for sample type airborne particulates...")

Technical corrections

P2 L4: "...the rest being associated..." instead of "...being the rest associated..."

P3 L24: under or equipped, not both

P4 L15: "...since year 2002, but Poland made available samples for year 1986" should

probably be "...in 2002, but Poland made available measurements since 1986"?

P4 L31: "each other" instead of "each other's"

P5 L7: "It is" instead of "Itis"; "...attention to field..." instead of "...attention over field..."

P5 L8: "represents the best" instead of "represents best"

P5 L12: "in De Cort et al. (2004)" instead of "in (De Cort, et al., 2004)"

P5 L17: "...106 measurements..." is probably "...10ˆ6 measurements..."?

P5 L25: "1996" is probably "1986"?

P5 L26-27: "gradually lost" instead of "lost gradually"

P8 L9: "Povinec et al. (2003) analyse..." instead of "(Povinec et al., 2003) analyse..."

P8 L27: "(http://www.radioecology-exchange.org/content/monitoring-sr-90-and-cs-137-milk-finland)." instead of "(http://www.radioecology-exchange.org/content/monitoring-sr-90-and-cs-137-milk-finland respectively)"

P9 L15: Link does not work (browser message is: server IP address could not be found).

DOIs are missing in the References (P11 L18, P11 L23,...). The readers would also benefit if the authors provided URL's and/or DOI's of public reports in the References (e.g. De Cort et al., 2004)

References

Carlson, D. and Oda, T.: Editorial: Data publication – ESSD goals, practices and recommendations, Earth Syst. Sci. Data, 10, 2275-2278, https://doi.org/10.5194/essd-10-2275-2018, 2018. De Cort, M., Tollefsen, T., Marsano, A. & Gitzinger, C.: Environmental Radioactivity in the European Community 2004- 2006, http://dx.doi.org/10.2788/25616. 2004. De Cort, M.; Sangiorgi, M.; Hernandez Ceballos, M.A.; Vanzo, S.; Nweke, E.; Tognoli, P.V.; Tollefsen, T.: REM data bank - Years 1984-2006. European Commission, Joint Research Centre (JRC) [Dataset] doi:10.2905/jrc-10117-10024 PID: http://data.europa.eu/89h/jrc-10117-10024 . 2007

---

## Referee Comment (RC3) · Anonymous Referee #3 · 19 Feb 2019

This natural background is enhanced by nuclear accidents . . . . . .

It's better to explain, among others, that efficient dose for the population and workers is calculated considering the natural radioactivity background and excluding the artificial one. So in general it's better distinguish between natural background and increments from the same natural background.

On page 4 Maybe it should be spent few word on the type of scientific checks: considering that generally there are formats to be filled sent in various countries -it's difficult

to understand which kind of control it was done: which is the quality of the control.

OnPag.7 Airborne generally it's made a measurement after an hour and a half and it's waited the decay of the short-lived products of Radon, lead and bismuth.

Finally, for the figures, A part from the captions in line with the base of rectangle that contains them, I would suggest that - more than the progressive order generated by the date of membership of each country – starting from figure 3, it would be better an ascending or descending order, this order could be determined by the number of measurements carried out by each country; even if a country has started after years, this country could be able to take a number of measurements greater than those countries who have taken part from the beginning. (As in Figure 4, and Figure 2 at Pag18). However I point out that there are two figures

---

## Referee Comment (RC4) · Anonymous Referee #4 · 27 Feb 2019

General Comments: Long-term (30 years) environmental radiation monitoring datasets at the large regional scale (Europe) are described in detail. The data are interesting and valuable for the general public and scientific community. This paper is well written and suitable to be published in Earth System Science Data. In the following lines, authors will find minor comments: Page 5 Line 7: "Itis" should be "It is". Line 18: "106 measurements" should be "106 measurements". Line 19: "Surface water and Drinking water" seems not reasonable category. Page 15 Figure 2: "E+0" is 100? If so, Figure 2, 8, 9, 10, 11 should be same with Figure 3 & 5. Page 16 Table 1: "Altitude" is more

suitable than "Height". Page 17 Figure 4: What's the meaning of "logarithmic scale"? Is $1\times10^7$ a logarithmic value? In general, plants include trees, grass, moss, etc. In the sample category of "Trees, Plants, Moss, Grass", "Plants" specifically refer to what?

---

## Referee Comment (RC5) · Anonymous Referee #5 · 28 Feb 2019

General comment:

'30 years of European Commission Radioactivity Environmental Monitoring Database (REMdb) – an open door to boost environmental radioactivity research' document contain useful information for people interested to environmental radioactivity research and can be considered for publication.

Comments:

P2 L10-11 Please, consider if to remove/rephrase the last sentence (Nevertheless...).

[Figure]

P3 L13 'seem reasonably stable' do you mean that this is a complete data sample from each MS?

P3 L28 Please, describe in the text the flow shown in figure 1.

P4 L18-19 Even considering... The wide variability of the number of measurements per country could be due to a different number of measurement sites, a different area of the countries, specific country properties, etc. is it correct? If yes, I suggest changing the sentence taking it into account.

P4 L25 it could be useful to specify the main checks.

P5 L 12-33 it might be useful, to understand the power of the database, to mention the total number of variables currently available.

P7 L7 Please, consider if to change the subsection title with Air measurements.

P7 L25 Please, consider if to change the subsection title with Water measurements.

P8 L22 Please, consider if to change the subsection title with Milk measurements.

P9 L1 Idem.

P9 Section 4 Is it necessary to list all the files? I suggest to change the list with a sentence.

Minor comments:

P1 L28 and P2 L18 ionising-> ionizing.

P3 L31 A dot is needed at the end.

P4 L22 organisation-> organization.

P5 L17 sample type -> sample category.

References:

references are not homogeneously reported

Figures:

General comment, please, make the fonts size homogeneous.

Fig. 2 I suggest to remove the sentence in parenthesis (sentence already mentioned in the text) and add a dot to the end of the caption.

Fig.3 I suggest to remove the sentence in parenthesis and add the dot.

Table 1 Is it 'less than' mandatory? (see the text P5 L8).

Fig. 7 Add a dot at the end of the caption.

Table 2 Please, move Radionuclide category at the center of the field and add a dot.

Fig 8-11 The figures are not clearly legible.

Figs 9-11 please, change 'as recommended in Table 2' with 'recommended in (Basic Safety Standards, 2014)'.

---

## Referee Comment (RC6) · ARUNKUMAR ANBU ARAVAZHI (Referee) · 11 Mar 2019

11th March 2019 Dear Authors, Thanks for the manuscript (MS) essd-2018-160 on "30 years of European Commission Radioactivity Environmental Monitoring Database (REMdb) – an open door to boost environmental radioactivity research". I always found this work very relevant and ground-breaking in a way. The paper is well written and it deserves to be published, especially as the existence of REMdb is not that well known to all. Even after 30 years work experience with environmental radioactivity, had never heard of such a resource. I suggest publication of the manuscript in Earth System

[Figure]

Science Data. However, the text still suffers from spacing problems, namely, the space between the values and its units. Please provide those updates and be very thorough! Below I provide an incomplete list of consistent problems in many phrases. Once authors have completed those very relevant details to satisfaction we can move ahead with a final check for publication. I hope we could reach that level for such an important topic indeed. Page 3, L17: The abstract says the database contains measurements since 1984, while in page 3 it says since 1988, REMdb was set-up in 1988 explain. Page 5, line 24: "In 1996, during the Chernobyl accident, there was. . . . . .:" Chernobyl accident took place at 1986? Kindly clear it. Page 9 L15: Link does not work (browser message is: server IP address could not be found). Kindly verify.

Fig. 7 shows the sampling distribution from 13 years ago. As the authors present the database up until 2016, a more recent picture would be appropriate.

References kindly follow the journal format.

Thanks once more to the authors and more then with their replies and update
* * *

---

## Author Comment (AC1) · 11 Mar 2019

Dear reviewer, thanks for the time you dedicated for reviewing this paper and all your good suggestions.

Every Member State competent authorities also publish at national level the same data submitted to REMdb; our database is a collection of all MSs' submissions. They are our only data providers. We don't have any plan of extending the time period backwards from 1984.

[Figure]

Since we have received plenty of comments by five different reviewers, it is difficult to reply one by one in this window; please have a look at the revised paper which takes into account all of them.

For some of them, please find below our answer.

Is the high Cs-137 content of ocean water in the Irish Sea due to Sellafield emissions or the Chernobyl accident? We don't know, we are not in charge of giving an answer, our role is to collect and compile data; local national authorities are in charge to give explanations.

Maybe the gaseous iodine should also be discussed. Because of the short decay times of iodine radionuclides (I-131:($T\frac{1}{2}$)= 193h , Te-132/I-132:($T\frac{1}{2}$)= 77.5h, I-133:($T\frac{1}{2}$)= 20.9h, I-134:($T\frac{1}{2}$)= 0.875h, I-135:($T\frac{1}{2}$)= 6.57h) the presence of this kind of nuclides is strictly related to the time of release / accident. Gaseous iodine requires specific sampling procedure that are not applied routinely from the most of laboratories. In REMdb's list of nuclides "I-131(G):IODINE-131 GAS" is included, therefore it would be possible to add these measurements if available.

How about "after the decay of short-lived radon progeny"? The short lived radon daughters are excluded through a sufficient delay time (e.g. five days) before counting (2000/473/Euratom). In general tritium and very low energy beta emitters are normally not included in the total measurement activity (2000/473/Euratom). T-BETA-ART e T-ALFA-ART are present in REMdb list of nuclides. Most of the laboratories carry out both type of measurements for air samples.

Is beryllium-7 significant from dose point of view? If so, please, add a literature reference. We changed our text in the paper. Be7 is used because its presence in well detectable in the spectrometric measurement and its identification is a continuous check system for the users. "Beryllium-7 should be reported as a qualitative check of the methods used." (2000/473/Euratom)

[Figure]

In the meantime, we have released a new dataset including measurements from 2007 to 2011; we added it to the new paper draft.

Please also note the supplement to this comment: https://www.earth-syst-sci-data-discuss.net/essd-2018-160/essd-2018-160-AC1-supplement.pdf

———————————————————

---

## Author Comment (AC2) · 11 Mar 2019

Dear reviewer, thanks for the time you dedicated for reviewing this paper and all your good suggestions.

At the time the paper was sent for review, data from 2007 onward were not released and, therefore, not cited in the paper. In the meantime, we just released a brand new dataset with data from 2007 to 2011, added them in our JRC catalogue and assigned a DOI. Indeed we can only publish datasets whose monitoring reports are already

published, therefore up to 2011.

Because we have received plenty of comments by five different reviewers, it is difficult to reply to all of them one by one in this window; please have a look at the revised paper which takes into account them all.

A short description of the REMdb online query tool and its functionality was not included because we thought it would be out of the scope of this paper and, mainly, because we have plans to update it and change it very soon.

Newer data are also added to the new paper draft, please have a look.

Please also note the supplement to this comment:
https://www.earth-syst-sci-data-discuss.net/essd-2018-160/essd-2018-160-AC2-supplement.pdf

―――――――――――――――――――

[Figure]

**Supplement:**

[revised manuscript text omitted]

---

## Author Comment (AC3) · 11 Mar 2019

Dear reviewer, thanks for the time you dedicated for reviewing this paper and all your good suggestions.

We reviewed the paragraph about "natural background enhanced by nuclear accidents". –> As a consequence of anthropogenic activities, such as Chernobyl and Fukushima (e.g. Imanaka et al., 2015), detonations of nuclear weapons (e.g. Gabrieli et al., 2011), nuclear waste handling and disposal, medical procedures (e.g. diagnostic

[Figure]

X-rays, radiation therapy) (e.g. Alkhorayef et al., 2018) and mining (e.g. Carvalho et al., 2014), the background radioactivity level is increased.

About ordering alphabetically countries in Fig. 3, we think that we would loose information about the year a country joined REMdb. The other figures follow the same order for coherence.

In the meantime, we have released a new dataset including measurements from 2007 to 2011; we added it to the new paper draft. Please have a look at it.

Please also note the supplement to this comment:
https://www.earth-syst-sci-data-discuss.net/essd-2018-160/essd-2018-160-AC3-supplement.pdf

---

## Author Comment (AC4) · 11 Mar 2019

Dear reviewer, thanks for the time you dedicated for reviewing this paper and all your good suggestions.

We took into account all your suggestions and integrated them in a new paper version in which we have included a brand new dataset with data from 2007 to 2011.

Please have a look at the supplement material.

[Figure]

Regards

Please also note the supplement to this comment:
https://www.earth-syst-sci-data-discuss.net/essd-2018-160/essd-2018-160-AC4-supplement.pdf

[Figure]

**Supplement:**

[revised manuscript text omitted]

---

## Author Comment (AC5) · 11 Mar 2019

Dear reviewer, thanks for the time you dedicated for reviewing this paper and all your good suggestions.

We took into account all your suggestions and integrated them in a new paper version in which we have included a brand new dataset with data from 2007 to 2011. Please have a look at the supplement material.

Regarding the spelling, we have a doubt: we are using the U.K. English spell-check

and words such as ionising, organisation,.. are spelled with "s" and not with "z". We are not native English speakers, so we cannot have an opinion on that; we just use the word processor suggested spelling.

Please also note the supplement to this comment:
https://www.earth-syst-sci-data-discuss.net/essd-2018-160/essd-2018-160-AC5-supplement.pdf

**Supplement:**

[revised manuscript text omitted]

---

## Author Comment (AC6) · 11 Mar 2019

Dear reviewer, thanks for the time you dedicated for reviewing this paper and all your good suggestions.

Most of your comments were already addressed in the previous ones, please have a look to the new paper version in the supplement to this answer.

We will check the space between values and units, thanks.

[Figure]

Before datasets may be published, the corresponding monitoring reports must be issued first. At the time this paper is prepared, monitoring reports up to year 2011 were published; therefore it's possible to release datasets up to 2011. People are invited to check our data catalogue for updates.

Indeed, when we submitted the paper for review in January 2019, we could release data just up to 2006; in the meanwhile we could release data up to 2011 and we updated the manuscript accordingly. In REMdb we have data until very recent dates, but we cannot release them until the corresponding monitoring reports are issued.

Please also note the supplement to this comment:
https://www.earth-syst-sci-data-discuss.net/essd-2018-160/essd-2018-160-AC6-supplement.pdf

---

## Author Response (AR1)

Referee comments:

30 years of European Commission Radioactivity Environmental Monitoring Database (REMdb) –an open door to boost environmental radioactivity research by Marco Sangiorgi et al.

The manuscript deals with a database containing radioactivity data from environment, food chains etc. The database data flow relies on the EU member states' authorities that regularly send national data to the European Union. The paper is well written

and it deserves to be published, especially as the existence of REMdb is not that well known. Even I, after 30 years work experience with environmental radioactivity, had never heard of such a resource. I suggest publication of the manuscript in Earth System Science Data once the authors have taken into consideration some minor suggestions found below.

General comments

To put the REMdb to a wider context I wonder if similar more or less public databases are available elsewhere? Are the MS competent authorities the only data providers? University datasets often provide useful information and nowadays the funding organizations often require an open data policy. Are there plans to extend the time period backwards from 1984? Important data was gathered during the period of atmospheric weapons testing.

Detailed comments

Citation on page 2, line 23: Do the authors mean the International Atomic Energy Agency & World Health Organization. (‎1996)‎. International basic safety standards for protection against ionizing radiation and for the safety of radiation sources. Vienna : International Atomic Energy Agency. http://www.who.int/iris/handle/10665/41593

Page 5, line 24: "In 1996, during the Chernobyl accident, there was…" 1986?

Page 6, line 31: "In fact, gross beta analysis does not detect weak beta-emitters such as those emitted by 3H, 14C, 35S and 129I." Maybe the authors should tell that total beta activity results are always dependent on the instrument used. Some instruments can measure even low-energy beta particles.

Page 7, lines 8-12: Maybe the gaseous iodine should also be discussed.

Page 7, Lines 13-14: "In most countries filters are changed daily and analysed for total beta activity following the decay of radon decay products." How about "after the decay

of short-lived radon progeny.

Page 7, lines 17-19: " 137Cs and 7Be are normally measured with a gamma spectrography at the same time, therefore the amount of reported measurements for both nuclide should be the same, but it does not happen because of lack of harmonization between countries." spectrography -> spectrometry?  both nuclide -> both nuclides?  Maybe the amount of reported measurements for both nuclides differ also due to "<MDA" values?

Page 7, lines 21-22. Is beryllium-7 significant from dose point of view? If so, please, add a literature reference.

Page 7, lines 25-31: Please, clarify the term "surface water". Does it mean fresh water in lakes and rivers or is also surface water of oceans included? I would expect the radionuclide content of water and intake by drinking to be negligible compared to aquatic food chains ending to man.

Page 8, lines 9-11. Is the high Cs-137 content of ocean water in the Irish Sea due to Sellafield emissions or the Chernobyl accident?

Page 8, lines 20-21: "Eventual presence of 3H, 90Sr and 137Cs and radium may also be due to man's activities." Isn't the presence of Sr-90 and Cs-137 solely due to anthropogenic activities?

[Figure]

Earth Syst. Sci. Data Discuss.,
https://doi.org/10.5194/essd-2018-160-RC2, 2019
The manuscript "30 years of European Commission Radioactivity Environmental Monitoring Database (REMdb) – an open door to boost environmental radioactivity research" describes the REMdb database which is a product of a more than a three decade-long radioactivity monitoring effort and collaboration of European member states. The long time span, vast geographical coverage, variety of sample types and the immense number of measurement records result in an invaluable dataset, which will undoubtedly prove of great value for the scientific community. In this light, the

manuscript fits very well into the scope of the journal "Earth System Science Data" and can be considered for publication after the authors address the comments posted below.

The manuscript provides links to yearly and bulk datasets which can be downloaded as Excel files. Data from REMdb can also be accessed by an online query tool where the user can personalise the search by location, sample type, observation period, export format etc. The files on the provided links and the files provided by the online query tool are compliant with the descriptions provided in the Data Availability section.

The manuscript accompanying data does, however, have a major issue which the authors should discuss with the Editor before revision. The present database is composed of two datasets. While the first one spanning between 1984-2006 (De Cort et al., 2007) is compliant with the data policies posted on ESSD websites and further elaborated in a recent Editorial (Carlson and Oda, 2018), the second dataset (2007-2016) is not. Namely, it does not have a DOI nor is it fully publicly available (explicit request by email is needed for access; P10 L18). Additionally, the part of the Disclaimer in P11 L10-11 ("The European Union reserves the right to . . . discontinue temporarily or permanently, the REM Database. . .") could prove controversial regarding the above mentioned data policies of ESSD.

Specific comments

In P1 L9 the DG abbreviation is not explained.

P3 L10 and P1 L17: The abstract says the database contains measurements since 1984, while in page 3 it says since 1988.

P7 L15 and Fig. 8: "Figure 8 shows the amount of measurements by country for 137Cs and 715 Be in the air." For unambiguity the authors should clarify that this refers to the total amount of measurements in the database.

P7 L26: "aquatic" is probably more appropriate than "marine"?

[Figure]

Section 4: The "Data availability" section should include procedure for data after 2006, i.e. it should be explicitly stated in P10 L18 that the full database also contains measurements after 2006. Additionally, I suggest the authors do not only mention, but also include a short description of the REMdb online query tool and its functionality as it offers useful search options and additional export formats which many readers could find beneficial.

P10 L11-14: The abbreviations used in the Excel files should be mentioned in the paper, for example: "locality name (LOC_NAME),..., apparatus description (APT_DESCRIPTION), nuclide (NUC_CODE)..."

Figure design of the graphs in the manuscript is variable, for example: some have a frame (Figs. 2, 8-11) and some do not (Figs. 3-6); font sizes of axis titles in Figs. 5 and 6 are much larger compared to similar graphs in the manuscript.

Fig. 7 shows the sampling distribution from 13 years ago. As the authors present the database up until 2016, a more recent picture would be appropriate.

Fig. 8: The legend in the figure is so small that the reader cannot see which symbols are used for 137Cs, total beta and 7Be

Fig. 9a: Again the legend is too small to recognise the symbols of the radionuclides

Figs. 9a and 9b: There should be only one subscript per figure

Figs. 8-11: I suggest to add "in REMdb" to avoid ambiguity (e.g. "Total amount of measurements in REMdb (dense network) for sample type airborne particulates...")

Technical corrections

P2 L4: "...the rest being associated..." instead of "...being the rest associated..."

P3 L24: under or equipped, not both

P4 L15: "...since year 2002, but Poland made available samples for year 1986" should

probably be "...in 2002, but Poland made available measurements since 1986"?

P4 L31: "each other" instead of "each other's"

P5 L7: "It is" instead of "Itis"; "...attention to field..." instead of "...attention over field..."

P5 L8: "represents the best" instead of "represents best"

P5 L12: "in De Cort et al. (2004)" instead of "in (De Cort, et al., 2004)"

P5 L17: "...106 measurements..." is probably "...10ˆ6 measurements..."?

P5 L25: "1996" is probably "1986"

P5 L26-27: "gradually lost" instead of "lost gradually"

P8 L9: "Povinec et al. (2003) analyse..." instead of "(Povinec et al., 2003) analyse..."

P8 L27: "(http://www.radioecology-exchange.org/content/monitoring-sr-90-and-cs-137-milk-finland)." instead of "(http://www.radioecology-exchange.org/content/monitoring-sr-90-and-cs-137-milk-finland respectively)"

P9 L15: Link does not work (browser message is: server IP address could not be found).

DOIs are missing in the References (P11 L18, P11 L23,...). The readers would also benefit if the authors provided URL's and/or DOI's of public reports in the References (e.g. De Cort et al., 2004)

[Figure]

It's better to explain, among others, that efficient dose for the population and workers is calculated considering the natural radioactivity background and excluding the artificial one. So in general it's better distinguish between natural background and increments from the same natural background.

On page 4 Maybe it should be spent few word on the type of scientific checks:
[Figure]
 considering that generally there are formats to be filled sent in various countries -it's difficult

to understand which kind of control it was done: which is the quality of the control.

OnPag.7 Airborne generally it's made a measurement after an hour and a half and it's waited the decay of the short-lived products of Radon, lead and bismuth.

Finally, for the figures, A part from the captions in line with the base of rectangle that contains them, I would suggest that - more than the progressive order generated by the date of membership of each country – starting from figure 3, it would be better an ascending or descending order, this order could be determined by the number of measurements carried out by each country; even if a country has started after years, this country could be able to take a number of measurements greater than those countries who have taken part from the beginning. (As in Figure 4, and Figure 2 at Pag18). However I point out that there are two figures
* * *
[Figure]

Earth Syst. Sci. Data Discuss.,
https://doi.org/10.5194/essd-2018-160-RC4, 2019

[Figure]

General Comments: Long-term (30 years) environmental radiation monitoring datasets at the large regional scale (Europe) are described in detail. The data are interesting and valuable for the general public and scientific community. This paper is well written and suitable to be published in Earth System Science Data. In the following lines, authors will find minor comments: Page 5 Line 7: "Itis" should be "It is". Line 18: "106 measurements" should be "106 measurements". Line 19: "Surface water and Drinking water" seems not reasonable category. Page 15 Figure 2: "E+0" is 1.0? If so, Figure 2, 8, 9, 10, 11 should be same with Figure 3 & 5. Page 16 Table 1: "Altitude" is more

suitable than "Height". Page 17 Figure 4: What's the meaning of "logarithmic scale"? Is $1\times10^7$ a logarithmic value? In general, plants include trees, grass, moss, etc. In the sample category of "Trees, Plants, Moss, Grass", "Plants" specifically refer to what?

[Figure]

Earth Syst. Sci. Data Discuss.,
https://doi.org/10.5194/essd-2018-160-RC5, 2019

'30 years of European Commission Radioactivity Environmental Monitoring Database (REMdb) – an open door to boost environmental radioactivity research' document contain useful information for people interested to environmental radioactivity research and can be considered for publication.

Comments:

P2 L10-11 Please, consider if to remove/rephrase the last sentence (Nevertheless...).
[Figure]

[Figure]

P3 L13 'seem reasonably stable' do you mean that this is a complete data sample from each MS?

P3 L28 Please, describe in the text the flow shown in figure 1
[Figure]

P4 L18-19 Even considering... The wide variability of the number of measurements per country could be due to a different number of measurement sites, a different area of the countries, specific country properties, etc. is it correct? If yes, I suggest changing the sentence taking it into account.

P4 L25 it could be useful to specify the main checks.

P5 L 12-33 it might be useful, to understand the power of the database, to mention the total number of variables currently available.

P7 L7 Please, consider if to change the subsection title with Air measurements.

P7 L25 Please, consider if to change the subsection title with Water measurements.

P8 L22 Please, consider if to change the subsection title with Milk measurements.

P9 L1 Idem.

P9 Section 4 Is it necessary to list all the files? I suggest to change the list with a sentence.

Minor comments:

P1 L28 and P2 L18 ionising-> ionizing.

P3 L31 A dot is needed at the end.

P4 L22 organisation-> organization.

P5 L17 sample type -> sample category.

References:

[Figure]

references are not homogeneously reported

Figures:

General comment, please, make the fonts size homogeneous.

Fig. 2 I suggest to remove the sentence in parenthesis (sentence already mentioned in the text) and add a dot to the end of the caption.

Fig.3 I suggest to remove the sentence in parenthesis and add the dot.

Table 1 Is it 'less than' mandatory? (see the text P5 L8).

Fig. 7 Add a dot at the end of the caption.

Table 2 Please, move Radionuclide category at the center of the field and add a do

Fig 8-11 The figures are not clearly legible.

Figs 9-11 please, change 'as recommended in Table 2' with 'recommended in (Basic Safety Standards, 2014)'.
* * *
[Figure]

Earth Syst. Sci. Data Discuss.,
https://doi.org/10.5194/essd-2018-160-RC6, 2019

11th March 2019 Dear Authors, Thanks for the manuscript (MS) essd-2018-160 on "30 years of European Commission Radioactivity Environmental Monitoring Database (REMdb) – an open door to boost environmental radioactivity research". I always found this work very relevant and ground-breaking in a way. The paper is well written and it deserves to be published, especially as the existence of REMdb is not that well known to all. Even after 30 years work experience with environmental radioactivity, had never heard of such a resource. I suggest publication of the manuscript in Earth System

[Figure]

Science Data. However, the text still suffers from spacing problems, namely, the space between the values and its units. Please provide those updates and be very thorough! Below I provide an incomplete list of consistent problems in many phrases. Once authors have completed those very relevant details to satisfaction we can move ahead with a final check for publication. I hope we could reach that level for such an important topic indeed. Page 3, L17: The abstract says the database contains measurements since 1984, while in page 3 it says since 1988, REMdb was set-up in 1988 explain. Page 5, line 24: "In 1996, during the Chernobyl accident, there was. . . . . . :" Chernobyl accident took place at 1986? Kindly clear it. Page 9 L15: Link does not work (browser message is: server IP address could not be found). Kindly verify.

Fig. 7 shows the sampling distribution from 13 years ago. As the authors present the database up until 2016, a more recent picture would be appropriate.

References kindly follow the journal format

Thanks once more to the authors and more then with their replies and update

[revised manuscript text omitted]